# Animal source food consumption practice and factors associated among infant and young children from selected rural districts in Ethiopia: A cross-sectional study

**Alemneh Kabeta Daba** [ORCID] [1] *, **Mary Murimi** [2], **Kebede Abegaz** [3], **Dejene Hailu** [4]

1 School of Nursing, College of Medicine and Health Sciences, Hawassa University, Hawassa, Ethiopia, 2 Department of Nutritional Sciences, College of Human Sciences, Texas Tech University, Lubbock, Texas, United States of America, 3 School of Nutrition and Food Science and Technology, College of Agriculture, Hawassa University, Hawassa, Ethiopia, 4 School of Public Health, College of Medicine and Health Sciences, Hawassa University, Hawassa, Ethiopia

* alemnehkabeta@hu.edu.et

## Abstract

### Background

Children are recommended to consume animal source foods (ASF) as part of diversified diets. However, ASF consumption practice of infant and young children (IYC) is less studied and contributing factors are not exhaustively identified. Therefore, the purpose of this study is to assess consumption of ASF and identify associated factors among 6–23 months old IYC from selected rural districts in Ethiopia.

### Methods

A cross-sectional study was conducted in a total of 606 IYC from selected rural districts of Oromia and Sidama regional states in Ethiopia. A two-stage sampling technique was used to select participants. Data were collected using interviewer-administered questionnaire. Frequencies, percentages and mean scores with standard deviations were generated to describe participants and report univariate outcomes. Consumption of ASF was assessed using a 24-hour dietary recall. Logistic regression analysis was applied to identify contributing factors contributed to IYC's ASF consumption practice.

### Results

Dairy, eggs and meat were consumed by 41.2%, 16.4% and 2.3% of IYC, respectively. Household food security increased odds of dairy [AOR = 1.66 (95%CI: 1.16 2.38), P = 0.006], eggs [AOR = 2.15 (95%CI: 1.33, 3.49), P = 0.002] and meat [AOR = 5.08 (95%CI: 1.09, 23.71), P = 0.039] consumption. Cow [AOR = 1.86 (95%CI: 1.28, 2.70), P = 0.001], donkey [AOR = 1.83 (95%CI: 1.08, 3.11), P = 0.024] and chicken [AOR = 1.53 (95%CI: 1.05, 2.22), P = 0.027] ownership increased the odds of dairy consumption. Grades 5–8 [AOR = 1.74 (95%CI: 1.06, 2.86), P = 0.028] or ≥9 [AOR = 2.96 (95%CI: 1.62, 5.42), P

**Funding:** This work was funded by the United States Agency for International Development (USAID) Bureau for Food Security under Agreement No. AID-OAA-L-15-00003. The funder had no role in study design, data collection and analysis, decisions to publish, or preparation of the manuscript.

**Competing interests:** The authors have declared that no competing interests exist.

<0.001] maternal educational achievements were also associated with better dairy consumption. Children from households that owned chicken [AOR = 3.20 (95%CI: 1.97, 5.19), P <0.001] or produce root crops [AOR = 1.67 (95%CI: 1.05, 2.66), P = 0.031] were with increased odds to consume eggs.

## Conclusions

Low proportion of children consumed ASF. Household food security, livestock ownership, household income, root crop production and maternal education contributed to ASF consumption. Nutrition sensitive agricultural extension activities aided by nutrition education should be considered and evaluated for their effect on IYC's ASF consumption practice.

## Introduction

The first two years of life are critical period during which organs, body systems and immunity develop and children start to learn the environment [1]. During this period, good nutrition-supported by appropriate behavioral interventions plays an important role to help children grow and develop to their full potential and have healthier foundation for the rest of their lives [2]. The period is a long portion of the first 1000 days of life, which is considered as a window of opportunity during which good nutrition practices has to continue, and corrective nutrition interventions has to be practiced to rectify nutritional derangements for those happened during pregnancy [3]. For children who do not meet the nutritional requirements, this period was found as a time that they lose much of their growth and developmental potential [4].

Growing children are fully dependent on mature individuals around them for nutritional and others cares, and interaction. To support nutrition literacy of the care providers in order to serve children to meet their nutritional requirement, infant and young child feeding (IYCF) guides [5–8] have been developed. The guides mainly focus on breast and complementary feeding practices during the first two years of life. As one of the complementary feeding recommendations, dietary diversity is a useful indicator of nutritional quality of complementary foods consumed by infants and young children (IYC) [9]. To satisfy the minimum dietary diversity (MDD) recommendation, IYC are expected to consume at least five of the eight food groups (breastmilk, cereals and roots, legumes, vitamin A rich fruits and vegetables, other fruits and vegetables, dairy products, eggs and flesh) in a day. Animal source foods share about 38% (three of the eight food groups) of the food groups used to determine whether IYC met MDD or not. But, more than 70% of IYC in Ethiopia did not met MDD and ASF were consistently reported as the commonly missed food groups [10–13]. Despite the less frequent inclusion of ASF in complementary foods of IYC, pertaining to researches' focus, consumption of ASF was less considered as primary outcome in the studies [14–20]. Hence, the purposes of this study are to assess ASF consumption practice and identify associated factors among IYC from selected rural districts in Ethiopia. The findings have potential to helping the realization of Ethiopian food and nutrition policy and strategy [21] through food-based approach that aimed at tackling malnutrition in all its forms.

## Materials and methods

### Study area, design and period

The study was conducted in Sidama and Oromia regional states of Ethiopia. Three districts (woreda) [Negelle Arsi, Wondo Gent and Dale] were purposively selected from the two regional states as they were implementation areas for a project that aimed to link cattle nutrition with human nutrition [22]. Negele Arsi is a district in Oromia regional state of Ethiopia. Negele Arsi is bordered in the south by Shashemene Zuria, on the southwest by Lake Shala, in the west by the Sidama region, in the north by Misraq Shewa, and on the east by the Arsi zone. The altitude of this woreda ranges from 1500 to 2300 meters above sea level. The estimated area of the woreda is 1,400.16 square kilometers. Data of the land in the woreda shows that 29.9% is cultivable land, 4.3% pasture, 5.2% forest, and the remaining 60.6% is considered swampy, degraded or unusable. According to the Central Statistical Agency of Ethiopia (CSA), this woreda has an estimated population of 198,307, of which 100,626 are men and 97,681 are women. About 21% (42,054) of its population are urban dwellers. The majority of the inhabitants were Muslim (68.86%), followed by Ethiopian Orthodox Christian (20.2%), Protestant (8.99%) and Catholic (1.04%). The largest ethnic groups in Arsi Negele are Oromo (85.92%), followed by Amhara (7.69%), Kambata (2.73%), and Sodo Gurage (1.08%). Oromiffa is widely (83.65%) spoken language, followed by Amharic (11.89%), and Kambatigna (2.44%). There were 33 Farmers Associations with 21,777 members and 12 Farmers Service Cooperatives with 11,430 members [23].

Dale and Wondo Genet districts are in Sidama regional states of Ethiopia. Wondo Genet bordered on the south by Malga, on the west by Awash Zuria, and on the northeastern by the Oromia Region. According to CSA-Ethiopia, the woreda has a total population of 155,715, of which 79,664 are men and 76,051 are women. About 15% (23,125) of its population are urban dwellers. The majority of the inhabitants were Protestant (83.26%) followed by Muslim (7.4%), Ethiopian Orthodox Christian (6.69%) and Catholic (1.68%) [24]. Dale is bordered on the south by Aleta Wendo and Chuko districts, on the west by Loka Abaya, on the northwest by Boricha, on the north by Shebedino, and on the east by Wensho. The elevation of this woreda varies from about 1200 to 3200 meters above the sea level. Report about land in Dale shows that 81.9% is cultivable, 2.7% forest, and the remaining 15.5% is considered swampy, degraded or unusable. Important cash crops for Dale include corn, barley, haricot beans, local varieties of cabbage, and sweet potatoes. Coffee is also an important cash crop in Dale, with 15.38 square kilometers planted with this crop. According to CSA-Ethiopia, the woreda has a total population of 242,658, of which 122,918 are men and 119,740 women. The majority of the inhabitants were Protestants (79.98%) followed by Ethiopian Orthodox Christian (8.04%), Muslim (4.69%), Catholic (3.46%) and traditional religions (1.3%). The four ethnic groups in Dale district were Sidama (91.29%), Amhara (3.98%), Oromom (1.16%), and Welayta (1.01%). A community-based cross-sectional study design was applied to conducted this research. This study was conducted from May to August 2018.

### Sample size determination and sampling procedure

Sample size was calculated using single population proportion formula [25] at 95% confidence level, 0.05 margin of error and design effect of 1.5. Dairy consumption prevalence of 60.7% (12) was used to decide the sample size. Though a 5% non-response rate was considered during sample size determination, it was not applied for data collection as no non-responder was encountered. A two-stage sampling technique was used to select the enumeration kebeles and IYC. At stage-one four kebeles (n = 12), the lowest unit in the administrative structure of

Ethiopia, were selected randomly from each of the study districts (n = 3) as enumeration areas. Stage-two also involved two activities. First, a house-to house census was conducted to develop a list of eligible households, households with IYC, for each kebele. Secondly, an equal proportional allocation was used to select households with IYC from each of the kebele. Accordingly, a total of 606 IYC (25 from each kebele) were selected through a systematic random sampling technique. The "K" value for each kebele was determined dividing the total number of eligible households by 25, number of IYC each kebele expected to contribute to the study sample size.

## Data collection

Data were collected through face-to-face interviewer administered questionnaire. All the respondents were mothers of the IYC. The data collection questionnaire consisted sections for household food insecurity [26], dietary diversity assessment tool focused on ASF consumption practice [27] and socio-demographic and economic characteristics of the IYC, mothers and households. Household food insecurity was assessed on a recall basis during thirty days before the date of data collection. Consumption of ASF was assessed as part of dietary diversity assessment over 24-hours period before the day of data collection. The tool was translated to local languages (Oromiffaa and Sidamu Afoo), tested and necessary amendments were made before it was used for the actual data collection. Nutrition and health professionals with a minimum of undergraduate degree collected the data. Data collectors attended three days training on basics of research methods, data collection techniques, research ethics and purpose of this study. Completeness, quality and consistency of data were checked by the investigators, who served as field supervisors, at spot during the data collection at the field.

## Data analysis

Data were checked for completeness, coded, entered into SPSS version 20 for windows and cleaned. Overall ASF consumption data were generated by summing up "0"-not consumed in the last 24-hours and "1"-consumed in the last 24-hours coding for flesh foods/meat, eggs and dairy consumption statuses. The summation results for overall ASF consumption ranged from zero to three. Zero resulted for IYC who consumed none of the ASF, 1 for IYC who consumed one of the ASF, 2 for IYC who consumed two of the ASF and 3 for IYC who consumed three of the ASF. Then, results of the summation for overall ASF consumption data were dichotomized into consumed at least one of the ASF ("coded with 1") and consumed none of the ASF ("coded with 0") during the 24-hours before the survey.

Descriptive statistical outputs (frequencies, proportion and mean/median scores with SD/IQR) were computed to describe the participants and generate results on ASF consumption practice. Bivariate logistic regression analysis was done to identify eligible variables to be included in multivariable logistic regression model to identify determinants of ASF consumption. Based on the results of bivariate logistic regression, variable with P. <0.25 [28] were included in multivariable logistic regression model. Goodness of fit test was assessed for data distribution. Multicollinearity was checked by evaluating if variance inflation factors (VIF) were greater than 10. Bivariate logistic regression analyses were conducted to identify variables eligible (with P. <0.25) for multivariable logistic regression. Multivariable forward stepwise logistic regression analysis was applied to identify factors associated with flesh foods, eggs, dairy and overall ASF consumption practice at statistical significance level of 0.05 or below.

## Ethics statement

Letter of ethical clearance with *Ref №: IRB/027/10* was issued after the research proposal was reviewed by Institutional Review Board (IRB) of Hawassa University-College of Medicine and

**Table 1. Frequency and percentage distribution of maternal and children characteristics (n = 605).**

| Maternal Characteristics | Categories | Frequency (N) | Percentage (%) |
|---|---|---|---|
| Education | No Formal Education | 119 | 19.7 |
| | Grades 1–4 | 159 | 26.3 |
| | Grades 5–8 | 231 | 38.2 |
| | ≥Grades 9 | 96 | 15.9 |
| Ethnicity | Sidama | 381 | 63.0 |
| | Oromo | 187 | 30.9 |
| | Wolayta | 15 | 2.5 |
| | Others | 22 | 3.6 |
| Religion | Protestant | 409 | 67.6 |
| | Muslim | 164 | 27.1 |
| | Orthodox | 23 | 3.8 |
| | Catholic | 9 | 1.5 |
| Age | ≤26 | 363 | 60.0 |
| | ≥27 | 242 | 40.0 |
| Occupation | Housemaker | 481 | 79.5 |
| | Merchant | 50 | 8.3 |
| | Farmer | 47 | 7.8 |
| | Others | 27 | 4.5 |
| **Children Characteristics** | | | |
| Sex | Female | 301 | 49.8 |
| | Male | 304 | 50.2 |
| Age in months | 6–11 | 241 | 39.8 |
| | 12–23 | 364 | 60.2 |

Health Sciences. Participation was on voluntary basis. Informed written consent and assent were obtained from mothers of the IYC.

## Results

### Maternal and children characteristics

Response rate of the study is 100%. An observation was excluded from analysis because of mixed response for ASF consumption practice. About two-thirds of the mothers attended school from grades one up to eight (64%), are Sidama (63%) by ethnicity and protestant (67.6%). Near to two-thirds (60%) of the mothers were ≤26 years old with average age of 25.6 (5). Four out of five (79.5%) mothers were housemakers. Half (50.2%) of the children were male, while near to two-thirds (60.2%) were twelve to twenty-three months old with average age of 13.9 (5.4) (Table 1).

### Household characteristics

Nearly all (93.6%) households were male-headed. Women were reported to decide on purchase of household daily need in more than half (58.2%) of the families. More than half of the households were had five or less family members with a median score of 5 (IQR: 4, 7). Half of the households (53.4%) had two or more under-five children. The estimated annual income for 65.5% of the households was ≤10,000 Ethiopian Birr (ETB)/$312 [at $32 exchange (ETB to USD) rate]. Half (52.4%) of the households were food insecure. Local market (63%), family milk cow and/or goat (32.4%) and relatives and neighbors (4.6%) were reported as the usual

**Table 2. Frequency and percentage distributions of household characteristics (n = 605).**

| Characteristics | Categories | Frequency (N) | Percentage (%) |
|---|---|---|---|
| Head of the Household | Husband | 566 | 93.6 |
| | Wife | 23 | 3.8 |
| | Other | 16 | 2.6 |
| Decision on Daily Household Needs made by; | Both Husband and Wife | 186 | 30.7 |
| | Wife | 352 | 58.2 |
| | Husband/Other | 67 | 11.1 |
| Family Size | ≤5 | 360 | 59.5 |
| | ≥6 | 245 | 40.5 |
| Number of Under-five Children | One | 282 | 46.6 |
| | Two or more | 323 | 53.4 |
| Estimated Annual income in Ethiopian Birr | ≤10000/$312 | 396 | 65.5 |
| | 10001–20000/$312–624 | 110 | 18.2 |
| | 20001–30000/$624–937 | 49 | 8.1 |
| | >30000/$937 | 50 | 8.3 |
| Food Security | Food Insecure | 317 | 52.4 |
| | Food Secure | 288 | 47.6 |
| Agriculture Land Ownership | ≤0.5 Hectare | 265 | 43.8 |
| | >0.5 Hectare | 340 | 56.2 |
| Crop Production: Produce- | Cereal | 433 | 71.6 |
| | Roots | 244 | 40.3 |
| | Legumes | 287 | 47.4 |
| | Vegetables | 343 | 56.7 |
| | Cash Crops | 399 | 66 |
| | Produce diversity of crops | 429 | 70.9 |
| Livestock Ownership | Oxen | 144 | 23.8 |
| | Cows | 257 | 42.5 |
| | Goats | 69 | 11.4 |
| | Sheep | 62 | 10.2 |
| | Donkey | 83 | 13.7 |
| | Chicken | 249 | 41.2 |

source of milk for the households. More than half (56.2%) of the households owned more than half hectare of agricultural land. Cereal (71.6%), cash crops (66%), vegetables (56.7%), legumes (47.4%) and roots (40.3%) were the major crops produced by the households. Cows (42.5%), chicken (41.2%), oxen (23.8%) and donkeys (13.7%) were livestock owned by the households (Table 2).

## Dairy products, eggs, flesh foods and overall ASF consumption practice of the IYC

Data on ASF consumption practice of the IYC was collected with the application of the standard dietary diversity assessment tool in which eight food groups are listed including selected ASF (dairy products, eggs and flesh foods) [27]. Consumption of dairy products, eggs, flesh foods was assessed for all of the IYC participated in this study over 24-hour period prior to the data collection day. Because the primary interest of the current investigation is about the consumption practice of ASF by IYC, results are focused on dairy products, eggs, flesh foods and overall ASF. Accordingly, dairy products were consumed by 41.2% of the IYC while 16.4%

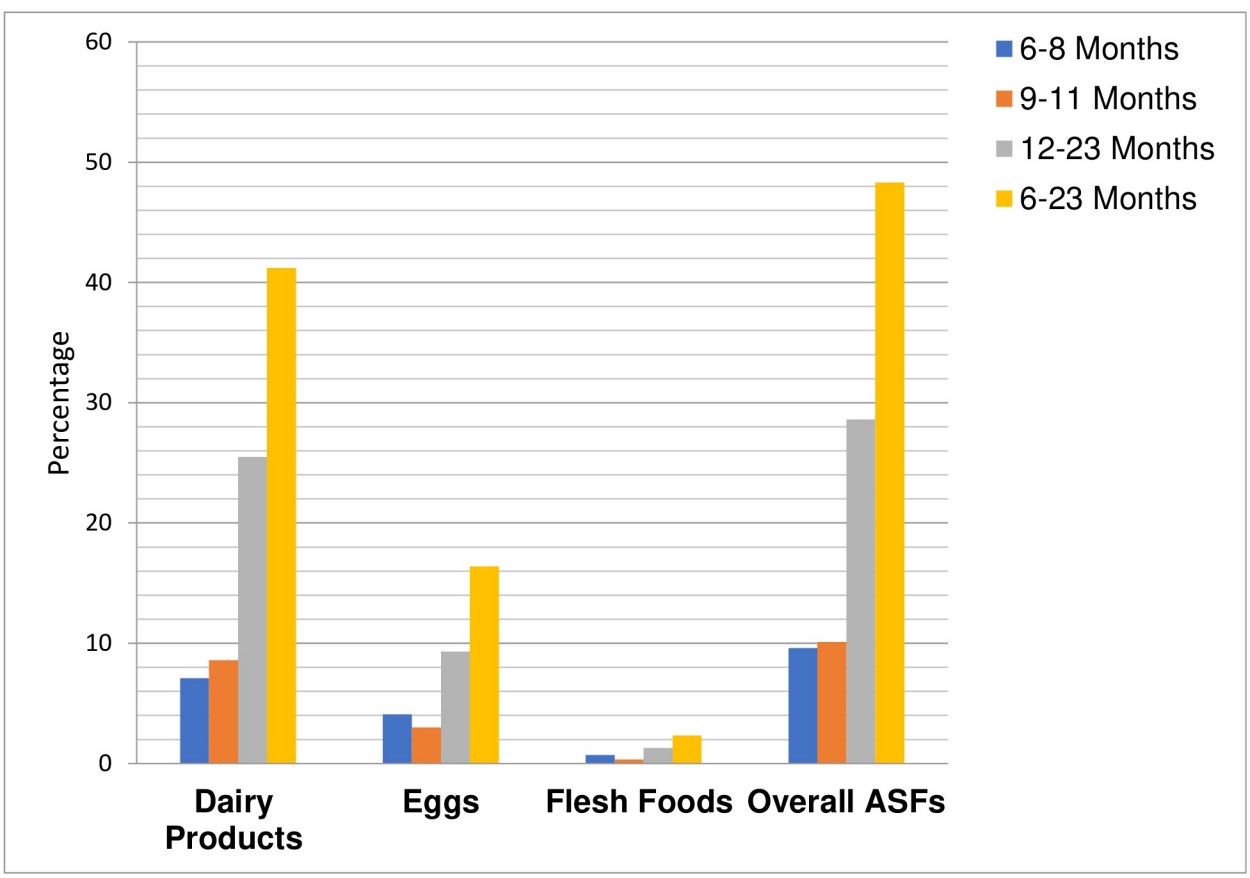

**Fig 1. Percentage distributions of children who consumed selected animal source foods, according to WHO recommended age category.**

consumed eggs and 2.3% consumed flesh foods. Collectively, less than half (48.3%) of the children consumed at least one of the three ASF in the past 24-hour (Fig 1).

### Factors associated with animal source foods consumption practice

Household food security was associated with higher odds of specific and overall ASF consumption. Infant and young children lived in food secure households were more likely to consume dairy [Adjusted Odds Ratio/AOR = 1.66 (95%CI: 1.16 2.38), P. = 0.006], flesh/meat [AOR = 5.08 (95%CI: 1.09, 23.71), P. = 0.039], eggs [AOR = 2.15 (95%CI: 1.33, 3.49), P. = 0.002] and at least one of the ASF [AOR = 1.94 (95%CI: 1.36, 2.78), P. <0.001] than IYC lived in food insecure households.

Educational status of the mothers and household-level livestock ownership was associated with higher odds of dairy consumption practice of IYC. Infant and young children lived in households that owned cow [AOR = 1.86 (95%CI: 1.28, 2.70), P. = 0.001] or chicken [AOR = 1.53 (95%CI: 1.05, 2.22), P. = 0.027] or donkey [AOR = 1.83 (95%CI: 1.08, 3.11), P. = 0.024] were about two times more likely to consume dairy than IYC who lived in households that did not own cow, chicken or donkey. Infant and young children with mothers who achieved school grades five up to eight were two times [AOR = 1.74 (95%CI: 1.06, 2.86), P. = 0.028], and IYC with mothers who achieved school grades nine or above had three times

higher odds [AOR = 2.96 (95%ci: 1.62, 5.42), P. <0.001] to consume dairy than IYC with mothers who never attended formal education.

Root crops production and chicken ownership were associated with increased eggs consumption. Infant and young children from households that owned chicken were three times [AOR = 3.20 (95%CI: 1.97, 5.19), P. <0.001] more likely to consume eggs than IYC from households that did not own chicken. Infant and young children who resided in households that produces root crops were also about two times [AOR = 1.67 (1.05, 2.66), P. 0.031] more likely to consume eggs than children in households that did not produce root crops.

Livestock ownership, household annual income and educational status of the mothers were associated with higher odds of overall ASF consumption practice. Infant and young children from households that owned cow [AOR = 1.89 (95%CI: 1.31, 2.74), P. = 0.001] or chicken [AOR = 1.93 (95%CI: 1.33, 2.81), P. = 0.001] were with odds of two to consume at least one of the ASF than IYC from households that did not own cow or chicken. Infant and young children from households with an estimated annual income of 10,001–20,000 [AOR = 1.70 (95% CI: 1.06, 2.73), P. = 0.028], 20,001–30,000 [AOR = 1.96 (95%CI: 1.01, 3.79), P. = 0.047] and more than 30,000 [AOR = 2.71 (95%CI: 1.36, 5.39), P. = 0.005] were with two to three increased odds to consume at least one of the ASF than IYC from households with an estimated annual income of 10,000ETB or less. Infant and young children with mothers who achieved school grades five up to eight were with odds of two [AOR = 2.03 (95%CI: 1.25, 3.31), P. = 0.004] to consume at least one of the ASF than IYC with mothers who never attended formal education (Table 3).

## Discussion

All children are in need of nutritional care that strengthen their healthy growth and development. They are recommended to be fed with diversified diets that include ASF [7, 29]. Animal source foods are beneficial for the growth and development of infant and young children [30, 31] as they are important sources of complete protein and a range of micronutrients [32–36]. In connection to this, the current study focused on the ASF consumption practice and its contributors among 6–23 months old IYC from selected rural districts in Ethiopia. To the result of this research, less than half (48.3%) of the children consumed at least one of the ASF (dairy or eggs or flesh foods). Dairy (41.2%) was consumed more frequently than eggs (16.4%) and flesh foods (2.3%). The proportion of IYC who consumed flesh is comparable with reports from Aleta Wondo-3.4% [37], Gorche-2.6% [12], Dangila-2.4% [11], Wollo-3.8% [38] districts in Ethiopia, but lower than proportions from Addis Ababa city-15.3% [39], Afar-29.3% [40], Bale-8.9% [41] and Wolaita Sodo-17.7% [42] towns in Ethiopia. The observed difference might be subjected to study setting differences as studies with higher proportion are conducted in a city or towns where ASF consumption enabling factors like availability and accessibility are better [43]. Eggs are consumed by lower proportion of IYC than study done in Wollo-6.6% [38], Dejen-98% [44] and Afar-10.2% [40] districts in Ethiopia. It was also consumed by a lesser proportion of IYC than children studied by studies from Woliata Sodo-31.6% [42], Aleta Wondo-21.5% [37], Addis Ababa-29.8% [39] and Bale-42% [41]. Research conducted in Gorche-11% [12], Sinan-11% [45] and Dangila-11.7% [11] districts of Ethiopia reported almost equal proportion of children who consumed eggs compared with proportion found by this study. Overall, the results indicated that animal source foods were lacked in complementary foods of considerable proportion of IYC. Consequently, animal source food centric IYC nutrition activities are highly demanding in the study areas to improve their inclusion in the complementary foods of children.

**Table 3. Factors associated with infant and young children's dairy products, flesh/meat, eggs and overall animal source foods consumption practice (n = 605).**

| Characteristics | Categories | AOR (95%CI) | *p.* |
|---|---|---|---|
| **Dairy Products[1]** | | | |
| Household Food Insecurity (Ref: Food Insecure) | Secure | 1.66 (1.16 2.38) | .006 |
| Cow Ownership (Ref: No) | Yes | 1.86 (1.28, 2.70) | .001 |
| Donkey Ownership (Ref: No) | Yes | 1.83 (1.08, 3.11) | .024 |
| Chicken Ownership (Ref: No) | Yes | 1.53 (1.05, 2.22) | .027 |
| Educational Status of the Mothers (Ref: No Formal Education) | Grades 5–8 | 1.74 (1.06, 2.86) | .028 |
| | $\geq$ Grades 9 | 2.96 (1.62, 5.42) | .000 |
| Household Estimated Annual Income in Ethiopian Birr (Ref: $\leq$ 10,000) | 20001–30000 | 2.22 (1.16, 4.26) | .017 |
| **Flesh/meat Foods[2]** | | | |
| Household Food Insecurity (Ref: Food Insecure) | Secure | 5.08 (1.09, 23.71) | .039 |
| **Eggs[3]** | | | |
| Household food Insecurity (Ref: Food Insecure) | Secure | 2.15 (1.33, 3.49) | .002 |
| Root Crops Production (Ref: No) | Yes | 1.67 (1.05, 2.66) | .031 |
| Chicken Ownership (Ref: No) | Yes | 3.20 (1.97, 5.19) | .000 |
| **Overall ASF[4]** | | | |
| Household Food Insecurity (Ref: Food Insecure) | Secure | 1.94 (1.36, 2.78) | .000 |
| Cow Ownership (Ref: No) | Yes | 1.89 (1.31, 2.74) | .001 |
| Chicken Ownership (Ref: No) | Yes | 1.93 (1.33, 2.81) | .001 |
| Estimated Household Annual Income in Ethiopian Birr (Ref: $\leq$ 10,000ETB) | 10001–20000 | 1.70 (1.06, 2.73) | .028 |
| | 20001–30000 | 1.96 (1.01, 3.79) | .047 |
| | >30000 | 2.71 (1.36, 5.39) | .005 |
| Educational Status of the Mothers (Ref: No Formal Education) | Grades 5–8 | 2.03 (1.25, 3.31) | .004 |

Variance Inflation Factors were < 10

Ref: Reference Category

[1/2/3/4]Variables in the multivariable forward stepwise logistic regression models to adjust for confounders: [[1]Dairy Products: Cow Ownership, Goat Ownership, Donkey Ownership, Chicken Ownership, Milk Source (Family Milk Cow/Goat or Local Market or Other), Household Food Insecurity, Roots Crop Production, Fruits Production, Cash Crops Production, Child Age in Months, Educational Status of the Mother, Maternal Occupation, Household Annual Income and Agriculture Land in Hectare; [2]Flesh/meat: Household food security, Sex of the child, Ox Ownership, Cereal Production Practice, Cow Ownership, Donkey Ownership, Family Size and Agricultural Land in Hectare; [3]Eggs: Root crops Production, Vegetable Production, Fruits Production Practice, Cash Crops Production Practice, Household Food Insecurity, Cow Ownership, Chicken Ownership, Educational Status of the Mother, Household Annual Income, Agricultural Land Size in Hectare; [4]Overall ASF consumption: Root Crops Production Practice, Cash Crops Production Practice, Mother got sick in the last two weeks, Household Food Insecurity, Ox Ownership, Cow Ownership, Goat Ownership, Sheep Ownership, Donkey Ownership, Chicken Ownership, Educational Status of the Mother, Household Annual Income, Maternal Occupation, Agricultural Land Size in Hectare and Head of the Household]

Household food security constantly increased probability of IYC's dairy, flesh and eggs consumption practice. The association could be explained by the incapability of food insecure households' purchasing power with the increasing cost of ASF [46], and the fact that food insecure households prefer to consume less expensive food products to cope with food insecurity [47]. The odds of dairy consumption practice were increased by cow, donkey and chicken ownership. The positive contribution of cow ownership can be connected with being a direct source of the food produce. On the other hand, in the study area as local market is the usual

source of dairy products for majority of the households (63%), donkey's role for income generation through charge for local transportation service might contributed to the positive association [48, 49]. Chickens might have also contributed to the increased consumption of dairy in a similar income generation pathway [50], in addition of being direct source of eggs and possibly flesh foods. Chicken themselves and commonly their food produce eggs can be sold at local markets and the cash can be used to purchase other food commodities. Similarly, odds of eggs consumption was higher for IYC from households that owned chicken. Better maternal educational achievement was also associated with more likelihood of dairy consumption. Our analysis also showed a positive effect of mother's higher educational achievement, cow or chicken ownership, better income category and household food security on improving the overall ASF consumption. The finding of a study done in Amhara-Ethiopia is in line with the observed association between cow or chicken ownership and overall ASF consumption in the present study [51]. Thus, in addition to the need to reduce burden of food insecurity, the move for inclusion of ASFs in complementary foods of IYC demands consideration of improving livestock husbandry, strengthening nutrition sensitive income pathway and maternal empowerment in nutrition literacy [52]. Further usage and application of findings of this study shall be in consideration of its curb in identifying the study districts, selection of enumeration areas/kebeles and portion of participant drown from the enumeration areas. In addition, the study used single-day dietary recall method to assess animal source food consumption practice.

## Conclusions

Majority of the IYC did not consume dairy products, eggs, flesh foods. Household food security was positively associated with all types of ASF and overall ASF consumption. Livestock (cow, chicken and donkey) ownership, better maternal educational achievement and better household income increased IYC's dairy consumption. Root crops production and chicken ownership were positively related to eggs consumption. Overall ASF consumption was determined by livestock ownership, household annual income and maternal education. In summary, consumption of ASF was dependent on socio-economic status and women empowerment as it was measured by household food security, livestock ownership, household income and maternal educational status. Agricultural extension activities to increase healthy productivity aided by nutrition education should be considered and evaluated for their effect on the IYC's ASF consumption practice in particular and diets in general. Women empowerment though education and income generation activities may also positively impact nutrition of IYC.

## Supporting information

**S1 Table. Table 4 bivariate logistic regression on dairy consumption of infant and young children.**
(DOCX)

**S2 Table. Table 5 bivariate logistic regression analysis on Flesh/Meat Consumption of infant and young children.**
(DOCX)

**S3 Table. Table 6 bivariate logistic regression analysis on eggs consumption of infant and young children.**
(DOCX)

**S4 Table. Table 7 bivariate logistic regression on overall ASF consumption of infant and young children.**
(DOCX)

**S1 Data.**
(SAV)

## Acknowledgments

The authors acknowledge the School of Nutrition, Food Science and Technology-Hawassa University, Livestock System Intensification Lab-Project and administrative bodies of the study districts for the assistance they provided throughout the study process.

## Author Contributions

**Conceptualization:** Alemneh Kabeta Daba, Mary Murimi, Kebede Abegaz, Dejene Hailu.

**Data curation:** Alemneh Kabeta Daba.

**Formal analysis:** Alemneh Kabeta Daba.

**Investigation:** Alemneh Kabeta Daba.

**Methodology:** Alemneh Kabeta Daba.

**Supervision:** Kebede Abegaz, Dejene Hailu.

**Writing – original draft:** Alemneh Kabeta Daba.

**Writing – review & editing:** Alemneh Kabeta Daba, Mary Murimi, Kebede Abegaz, Dejene Hailu.

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
