## [Decision Letter · Decision Letter 0]

2 Mar 2023

PONE-D-22-30710Predictors of Animal Source Food Consumption among Infant and Young Children from Rural Districts of Ethiopia: A Cross-sectional StudyPLOS ONE

Dear Dr. Daba,

Thank you for submitting your manuscript to PLOS ONE. After careful consideration, we feel that it has merit but does not fully meet PLOS ONE’s publication criteria as it currently stands. Therefore, we invite you to submit a revised version of the manuscript that addresses the points raised during the review process.

We look forward to receiving your revised manuscript.

Kind regards,

Tadele Tolosa Fulasa, Ph.D

Academic Editor

PLOS ONE

Journal Requirements:

2. In the ethics statement in the Methods, you have specified that verbal consent was obtained. Please provide additional details regarding how this consent was documented and witnessed, and state whether this was approved by the IRB.

Reviewers' comments:

Reviewer's Responses to Questions

**Comments to the Author**

1. Is the manuscript technically sound, and do the data support the conclusions?

Reviewer #1: Yes

Reviewer #2: No

2. Has the statistical analysis been performed appropriately and rigorously? 

Reviewer #1: Yes

Reviewer #2: No

3. Have the authors made all data underlying the findings in their manuscript fully available?

Reviewer #1: Yes

Reviewer #2: Yes

4. Is the manuscript presented in an intelligible fashion and written in standard English?

Reviewer #1: Yes

Reviewer #2: No

5. Review Comments to the Author

Reviewer #1: Why do you prefer single proportion formula while you are assessing the predictors/levels of significance/

The reason for selection of three dstrict (Why purposive)? can we generate an evidence on the selected study population to the general remaining population?

About Ethical approval of two different regional administration? Who should provide Ethical approval? IRB or National Research Ethical Review Committee? Nothing displayed (Reference number??)

Does your research Ethical approval from Oromia or Sidama?

Reviewer #2: It is an important research topic but the quality of the reporting and validity and reproducibility of methods employed to address the stated problem is unsatisfactory. The results and discussion sections need more improvement. Moreover, grammatical and editorial errors are frequent thought the document and thus the manuscript needs major revision.

6. PLOS authors have the option to publish the peer review history of their article (what does this mean?). If published, this will include your full peer review and any attached files.

Reviewer #1: No

Reviewer #2: **Yes: **Gebretsadik Berhe

---

## [Author Response · Author response to Decision Letter 0]

2 Jun 2023

Subject: Point-by-point response to Editors guidance and reviewers’ comments

Dear Editor,

Thank you for giving us the opportunity to submit a revised draft of the manuscript “Animal source food consumption practice and factors associated among infant and young children from selected rural districts in Ethiopia: A cross-sectional study” for publication in PLOSOne. We would like to thank the reviewers for their precious time and constructive comments. We have addressed the comments in a point- by- point response and a revised manuscript. The changes are highlighted in with track changes in the revised manuscript.

Journal Requirements:

Response: Thanks for the comment, and sharing link for formats and guidance. Changes are made accordingly. 

2. In the ethics statement in the Methods, you have specified that verbal consent was obtained. Please provide additional details regarding how this consent was documented and witnessed, and state whether this was approved by the IRB.

Response: Thanks for the feedback. Detailed Ethics statement is added to the methods and material subsection. 

Response: Data uploaded. 

Response: Detailed ethics statement is included in the methods section. 

Reviewer #1: 

Why do you prefer single proportion formula while you are assessing the predictors/levels of significance/

Response: Thanks for picking the point. Yes, it would have been good if sample size was calculated or the second section of the objective, predictor identification. However, to the level of literature search conducted before implementation of the current research there was no study that aimed to identify predictors of animal source food consumption among infant and young children. Thus, we were obliged to do the research using sample size computed with the application of single proportion formula. Luckily, the absence of literature on factors contributing to the inclusion of animal source foods in diets/complementary foods of infant and young children make this study a value adding article. 

The reason for selection of three dstrict (Why purposive)? can we generate an evidence on the selected study population to the general remaining population?

Response: Sure this is also an interesting pick! The districts were selected because they were areas where a project that aimed to link cattle nutrition with human nutrition was being implemented. They are selected purposively for this study, but randomly for the broader project that funded this analysis. Sure, we can generate inferable evidence as the districts were selected randomly by the broader project though this study used them purposively as it is part of the wider Nutrition Sensitive Agricultural project.

About Ethical approval of two different regional administration? Who should provide Ethical approval? IRB or National Research Ethical Review Committee? Nothing displayed (Reference number??)

Does your research Ethical approval from Oromia or Sidama?

Response: Good point! Description about research ethics was included in the online declaration assuming it will be included in the publication following acceptance of the manuscript. The research was reviewed and approved by Institutional Review Board of Hawassa University. Descriptions are included in the method subsection of the manuscript. 

Reviewer #2: 

It is an important research topic but the quality of the reporting and validity and reproducibility of methods employed to address the stated problem is unsatisfactory. The results and discussion sections need more improvement. Moreover, grammatical and editorial errors are frequent thought the document and thus the manuscript needs major revision.

Response: Thanks for echoing that the research topic is important. And that is why we tried to address it with all the limitation. Concerning validity and reproducibility of the methods employed we believe that it is to the level of epidemiological recommendations and nutritional assessment guidance. As no method and procedure to be used for research is free of drawbacks, the current research shall also shoulder this universal existence of limitations in methods and procedures to be applied for research execution. With all these, the study employed cross-sectional study design and followed standard sampling procedures that are described in the manuscript. Data was managed with acceptable statistical analyses and results were interpreted to the level best. Tool to collect data was compiled from recommended standard tools and literature review. Results and discussion section are revised. Grammatical and editorial errors are considered with the help of language experts. Overall, we accepted the recommended need for major revision and made it effective. 

Specific Comments

• Title: The title needs modification in light of the cross-sectional nature of the study design, authors can’t determine predictors of animal Sources Food Consumption among Infants and Young Children using a cross-sectional study design. For determining predictors authors need to use either a cohort or an interventional study design. By using a cross-sectional study, one can determine only the associated factors for the outcome variable. Moreover, the phrase “Rural Districts of Ethiopia” is very broad considering the large surface area of Ethiopia.

Response: Corrections are made as per the recommendation. To narrow the scope of the title in terms of geographic coverage included the word “selected.” 

Abstract: 

• In the background section of the abstract, the problem statement and rationale for conducting the study are missing. There is a miss-match between the title and the specific objectives described in the abstract. In the title, “Determination of predictors …” while in the objective “assess consumption of ASF and identify their predictors”.

Response: Good point! Yes it was lacking just to shorten the abstract as the information is well addresses in the background. We accepted the comment and included the research gap. Revision was made to manage the observed mismatch between title and objective. 

• In the method section: Authors failed to describe the specific study area, the sampling procedure, the type of data collected, who collected the data, and what type of descriptive analysis was carried out.

Response: Recommendation is accepted and information included. 

• Results Section: the authors used logistic regression analysis to identify associated factors with animal source food consumption. However, they use p values to report results. They should use AOR along with 95% CI as this help us to report the statistical significance, strength of association and direction of association.

Response: All right. It would have been good if AOR with 95%CI are included. But we only used p values to make the abstract brief and informative. Comment accepted and AOR with 95%CI included. 

Introduction: Authors need to explicitly describe the problem statement for their study. 

Response: Thanks for the comment and recommendation. Revision is made to explicitly indicate problem statement. At the end of the Introduction section of the manuscript it was described as “Despite the less inclusion if ASF in diets of IYC, pertaining to researches’ focus, assessment on the consumption of ASF and identifying contributing factors were less addressed as a primary outcome of the studies.” 

Methods:

• The study areas were not adequately described including population distribution, geography, socio-economy status and health infrastructure etc. 

Response: Comment accepted and study area description broadened. 

• On Page 4, 2nd paragraph, the authors used “Sampling” as a title for Sample Size Determination and Sampling Procedure. The word sampling is incomplete and thus should be replaced by Sample Size Determination and Sampling Procedure.

Response: recommendation accepted and correction made! 

• In sample size calculation, authors didn’t provide the expected prevalence assumed for calculation of the sample size and didn’t consider for non-response.

Response: Comment accepted and process followed described. 

• In the sampling procedure, authors said that they selected randomly four Kebelle from each district but they didn’t elaborate why they selected four Kebelles as number of Kebelles can vary from district to district. They didn’t provide also justification why they selected 25 study subjects from each Kebelle. There is no “Equal Proportional Allocation” sampling technique. It is not also clear how the systematic sampling method was conducted.

Response: Thanks for the observation and comment. Sure, the study is done as an extension of a developmental Nutrition Sensitive Agricultural intervention project, and aligning all academic requirements of conducting a research with such development works used to be challenging. However, we believe that we tried to address basics of sampling with all must to shoulder difficulties and also described the methodological recommendations in the discussion section of the manuscript as recommended. 

• On Page 4, 3rd paragraph, First sentence, Authors wrote that “Data were collected through one-on-one interviewer administered questionnaire”. What does “One-on-one” mean? Is it meant to say face-to-face?

Response: Good to pick such confusing narrations. It was to mean no mass data collection was made. By saying “One-on-one”, we meant each participant was treated separately for data collection. It is also an explanation for privacy related concerns. Comment is accepted and changed in to face-to-face. 

• Authors didn’t describe how they measured the independent and dependent variables. No operational definition was given. No information was given as to who collected the data.

Response: In the “Data Collection” sub section of the manuscript we have described how and using what primary outcome (Animal Source Food Consumption practice) was measured and appropriate citation are made. Accepted concern related with information on data collectors and included a sentence. 

• The data quality control used was inadequate. Did authors translate their tool to local language? Did they use supervision during data collection? Did they check completeness of the collected data on daily basis?

Response: Thanks for the detailed review, comment and recommendation. All the points raised were conducted during implementation of the study but missed in the narration. Corrections are made accordingly. 

• With regards to ethical consideration, authors claimed that “All methods were carried out in accordance with the Declaration of Helsinki”. This is vague, incomplete and inappropriate description. This should not be described under the “Data collection” heading. There is no information as to which University Institutional Review Board gave the permission? Dis authors use verbal or written informed consent? What did they do to ensure autonomy, beneficence and do not harm?

Response: Ethics statement added to the methods section. 

• In the data analysis section, authors failed to include multicollinearity test and goodness of fit test for the final logistic model. 

Response: Thanks for pointing out non reported assumption checks. Included. 

Results: 

• What is the response rate for the study? It needs to be stated.

Response: Included. 

• On page 6, Table 1: Second column should be labeled as categories; in Table 1 and Table 2, the column label for the third and fourth column should be revised as Frequency (N) and Percentage (%), respectively. What is “Age completed years” mean in Table 1? It is not clear and why it is categorized as < 26 and > 27.

Response: Frequency and percentage labeling made. Age in completed years is to mean that we have not considered months and not rounded for it. Changed in to the common “Age.” Concerning age categorization we use mean score (25.6Yrs) for age of our participants. 

• On Page 8, Table 2: The reason for categorizing family size, crop production and livestock ownership is not clear. Family size is categorized into <5 and <6. In the crop production and livestock ownership, what if a farmer produces two, three, four, five or all type of crops and a farmer owns two or more of the livestock? 

Response: family size categorization was in consideration of observed median score and average Ethiopian family size. The “<” sign before 6 is a typo error and replaced with “>.” Concerning categorization of crop production, data was collected about what crop/s a household used to produce and grouped in to the categories (Cereal, Legumes, Vegetable….) assuming nutrition classification. A Household was counted for more than one groups of crop if produced crops to be grouped in different crop categorization. That is why the results become more than the total number of participants. Results are reported only for producers. For example, in the result it is reported that n=433/71.6% of households produce cereals (maize, barely, sorghum, wheat…..). This means the remaining households (n=173) do not produce cereals. The report about livestock only ownership. Though data on numbers owned was collected there is no much difference in number of ownership. Thus, we dropped number and reported only ownership. 

• On Page 9: The y-axis of figure is not labeled

Response: It is labeled. 

• Page 11, Table 3: The outcome variable (animal source food consumption) frequency and percentage is missing in Table 3. It doesn’t include also the bivariate analysis results (COR along with the 95% CI). Why is donkey ownership included in Table 3 while Table is about animal source food consumption? How ownership of donkey is can be associated with the consumption of animal source foods? The bivariate and multivariable results should be presented in the same Table. It is not also clear why some variables and categories are missed in Table 3.

Response: Yes, it would have been good if frequency and percentage of animal source food consumption and bivariate analysis results (COR along with the 95%CI) are included in Table 3. However, the congestion they create 

---

## [Decision Letter · Decision Letter 1]

17 Jul 2023

PONE-D-22-30710R1Animal source food consumption practice and factors associated among infant and young children from selected rural districts in Ethiopia: A cross-sectional studyPLOS ONE

Dear Dr. Daba,

Thank you for submitting your manuscript to PLOS ONE. After careful consideration, we feel that it has merit but does not fully meet PLOS ONE’s publication criteria as it currently stands. Therefore, we invite you to submit a revised version of the manuscript that addresses the points raised during the review process

We look forward to receiving your revised manuscript.

Kind regards,

Tadele Tolosa Fulasa, Ph.D

Academic Editor

PLOS ONE

Journal Requirements:

Additional Editor Comments (if provided):

Dear Alemneh Kabeta Daba

We have now received the reviewer' comments on your article submitted to PLOSONE.  

The reviewer has advised that your manuscript should become suitable for publication in our journal after appropriate minor revisions. Please try to address all the comments forwarded by the reviewer. If you are able to address the reviewers' comments, which you can find highlighted yellow on the manuscript, I would like to invite you to revise and resubmit your manuscript. I ask that you respond to each reviewer comment by either outlining how the criticism was addressed in the revised manuscript or by providing a rebuttal to the criticism. This should be carried out in a point-by-point fashion.

To allow the editors and reviewers to easily assess your revised manuscript, we also ask that you upload a version of your manuscript highlighting any revisions made.

To submit your revised manuscript, please log in as an author at https://www.editorialmanager.com/pone/ and navigate to the "Submissions Needing Revision" folder under the Author Main Menu. Your revision due date is July 30, 2023.

If you need additional time to address the concerns that came up in the review process, please let us know so we can discuss a plan for moving your paper forward.

I look forward to receiving your revised manuscript.

Kind regards,

Tadele Tolosa Fulasa (PhD)

Plose one academic editor

Reviewers' comments:

Reviewer's Responses to Questions

**Comments to the Author**

1. If the authors have adequately addressed your comments raised in a previous round of review and you feel that this manuscript is now acceptable for publication, you may indicate that here to bypass the “Comments to the Author” section, enter your conflict of interest statement in the “Confidential to Editor” section, and submit your "Accept" recommendation.

Reviewer #3: (No Response)

2. Is the manuscript technically sound, and do the data support the conclusions?

Reviewer #3: Partly

3. Has the statistical analysis been performed appropriately and rigorously? 

Reviewer #3: Yes

4. Have the authors made all data underlying the findings in their manuscript fully available?

Reviewer #3: Yes

5. Is the manuscript presented in an intelligible fashion and written in standard English?

Reviewer #3: No

6. Review Comments to the Author

Reviewer #3: As a reviewer being invited to review this revised manuscript, my main concerns are listed as follow:

1. It seems the authors have not mentioned whether confounders have been adjusted in the regression analysis. This need to be mentioned.

2. Seemingly, determinants in the multivariate models could be correlated. I see the authors have investigated this, but it would be valuable for the authors to present their check on multicollinearity based on VIF.

3. The use of English can still be strengthened in the revised manuscript. My suggested edits are embedded in the attached manuscript.

Other comments and their corresponding areas (highlighted in yellow) needing the authors' attentions are listed in the attachment. For my answer of "partly" to question 2 above, please refer to my comments in Conclusions in the attachment.

7. PLOS authors have the option to publish the peer review history of their article (what does this mean?). If published, this will include your full peer review and any attached files.

Reviewer #3: **Yes: **Dehao Chen

---

## [Author Response · Author response to Decision Letter 1]

28 Nov 2023

Dear reviewer, we thank you for your valuable contributions. We learnt a lot from your views! Your contributions and comments helped us to strengthen our manuscript. It also provided us an opportunity to be more familiar with what we are communicating. THANKS A LOT! We have addressed the comments in a point- by- point response and a revised manuscript. The changes are highlighted with track changes in the revised manuscript.

Point-by-Point Responses; 

• Need for grammatical corrections throughout the manuscript 

Grammatical errors are addressed following recommendation form the reviewer and with the help of language expert. We would love also to thank the reviewer for making recommendation in the manuscript. 

• Abstract: Provide prevalence of ASF consumption 

Results are included in the results subsection of the abstract. 

• The need to update criteria for IYCF assessment 

Concepts and references are updated throughout the manuscript 

• Additional citations are needed 

More sources are cited in the manuscript including sources recommended by the reviewer. THANKS! 

• Data Collection section of the manuscript the reviewer requested the authors to articulate on how the recall bias was assessed for the sentence “Household food in/security was assessed on a recall basis during thirty days before the date of data collection.” 

We kindly request the reviewer to capture that the sentence is not about recall bias assessment. It is about how household food insecurity was assessed. This is to mean respondents we requested to recall their household-level food insecurity related experience in the past month, 30 days before the day of data collection. 

• Have the authors considered adjusting for confounders in the multivariable regression? 

Yes! The analysis was done two times (first bivariate and secondly multivariable) for each of the ASF and overall ASF consumption practice. Multivariable regression is one of the approaches with which we control for possible confounder during analysis. Results of the bivariate analysis are provided as a Supplementary Tables and variables considered in the multivariable regression model are listed as a foot note. 

• Seemingly, determinants in the multivariate models could be correlated. I see the authors have investigated this, but it would be valuable for the authors to present their check on multicollinearity based on VIF.

Yes, correlation between independent variables was assessed using variance inflation factor and it is described in the manuscript. Now, we have provided VIF related information on the foot note.

• Discussion section: The reviewer requested to further discuss the discussion we provided about the association between Donkey or Chicken ownership and dairy products consumption practice. 

Thanks for these value adding views so that our narrations are to the level of message we wanted to convey. Additional sentences are included to support readers understanding and avoid misinterpretation.

 With Best regards, 

Alemneh Kabeta Daba (PhD)

---

## [Decision Letter · Decision Letter 2]

15 Dec 2023

PONE-D-22-30710R2Animal source food consumption practice and factors associated among infant and young children from selected rural districts in Ethiopia: A cross-sectional studyPLOS ONE

Dear Dr. Daba,

Thank you for submitting your manuscript to PLOS ONE. After careful consideration, we feel that it has merit but does not fully meet PLOS ONE’s publication criteria as it currently stands. Therefore, we invite you to submit a revised version of the manuscript that addresses the points raised during the review process.

**Please address the minor comments raised by the reviewers and find the comments below.  ** Please submit your revised manuscript by Jan 29 2024 11:59PM. If you will need more time than this to complete your revisions, please reply to this message or contact the journal office at plosone@plos.org. Please include the following items when submitting your revised manuscript:A rebuttal letter that responds to each point raised by the academic editor and reviewer(s). You should upload this letter as a separate file labeled 'Response to Reviewers'.A marked-up copy of your manuscript that highlights changes made to the original version. You should upload this as a separate file labeled 'Revised Manuscript with Track Changes'.An unmarked version of your revised paper without tracked changes. You should upload this as a separate file labeled 'Manuscript'.If applicable, we recommend that you deposit your laboratory protocols in protocols.io to enhance the reproducibility of your results. Protocols.io assigns your protocol its own identifier (DOI) so that it can be cited independently in the future. For instructions see: https://journals.plos.org/plosone/s/submission-guidelines#loc-laboratory-protocols. Additionally, PLOS ONE offers an option for publishing peer-reviewed Lab Protocol articles, which describe protocols hosted on protocols.io. Read more information on sharing protocols at https://plos.org/protocols?utm_medium=editorial-email&utm_source=authorletters&utm_campaign=protocols.

We look forward to receiving your revised manuscript.

Kind regards,

Mohammed Feyisso Shaka, MPH

Academic Editor

PLOS ONE

Journal Requirements:

Reviewers' comments:

Reviewer's Responses to Questions

**Comments to the Author**

1. If the authors have adequately addressed your comments raised in a previous round of review and you feel that this manuscript is now acceptable for publication, you may indicate that here to bypass the “Comments to the Author” section, enter your conflict of interest statement in the “Confidential to Editor” section, and submit your "Accept" recommendation.

Reviewer #3: (No Response)

2. Is the manuscript technically sound, and do the data support the conclusions?

Reviewer #3: Partly

3. Has the statistical analysis been performed appropriately and rigorously? 

Reviewer #3: No

4. Have the authors made all data underlying the findings in their manuscript fully available?

Reviewer #3: (No Response)

5. Is the manuscript presented in an intelligible fashion and written in standard English?

Reviewer #3: (No Response)

6. Review Comments to the Author

Reviewer #3: While the authors have addressed many comments from the past iteration, it seems the results of regression analysis involving MDD have not been updated after the new cutoff is implemented - all the regression results are still the same as the past version. This needs to be addressed.

7. PLOS authors have the option to publish the peer review history of their article (what does this mean?). If published, this will include your full peer review and any attached files.

Reviewer #3: **Yes: **Dehao Chen

---

## [Author Response · Author response to Decision Letter 2]

14 Apr 2024

Date April 14, 2024

Subject: Point-by-point response to Editors guidance and reviewer’s comments

Dear Editor,

Thank you for giving us the opportunity to submit a revised version of the manuscript with title of “Animal source food consumption practice and factors associated among infant and young children from selected rural districts in Ethiopia: A cross-sectional study” for publication in PLOSOne. We would like to thank the reviewer for her/his precious time, constructive comments, teaching views and diligence for the betterment of our research output. We have addressed the comment in a point- by- point response below. This round we are not submitting revised manuscript with track changes as we only justified why we shall not revise the manuscript according to reviewer’s view.

Point-by-Point Response; 

• Reviewer #3: While the authors have addressed many comments from the past iteration, it seems the results of regression analysis involving MDD have not been updated after the new cutoff is implemented - all the regression results are still the same as the past version. This needs to be addressed. 

Dear Dehao Chen, thanks for your time, contribution, teaching views and diligence for the betterment of our research output. Let me/us use this as an opportunity that we enjoyed your comments and learnt a lot from your 360o views. Coming to the insightful comment provided….Yes, as the study period was May to Augest 2018, we used the revised (cited as [27]) standard dietary diversity assessment questionnaire for children under-two. However, from the data collected (eight food groups including breast milk) using the standard child dietary diversity assessment questionnaire, our being processed manuscript focused on selected animal source foods (Dairy products, Eggs and Flesh foods) consumption practice of children 6-24 months as outcome variables [Results Section-Dairy products, eggs, flesh foods and overall ASF consumption practice of the IYC section of the manuscript in the results sub-section], not MDD. Also, The logistic regression analyses were done to identify factors associated with IYC’s consumption practice of each of the selected animal source food and overall animal source food [Results-Section-Factors associated with animal source foods consumption practice & Table 3], not factors associated with MDD. To this end, though we revised the citation for source document of the data collection questionnaire we used for the outcome variables, we do not believe that the results from the regression analyses has to be updated. 

Kindly, 

Alemneh Kabeta Daba (PhD)

---

## [Decision Letter · Decision Letter 3]

21 Jun 2024

Animal source food consumption practice and factors associated among infant and young children from selected rural districts in Ethiopia: A cross-sectional study

PONE-D-22-30710R3

Dear Dr. Daba,

We’re pleased to inform you that your manuscript has been judged scientifically suitable for publication and will be formally accepted for publication once it meets all outstanding technical requirements.

Kind regards,

Mohammed Feyisso Shaka, MPH

Academic Editor

PLOS ONE

Additional Editor Comments (optional):

Reviewers' comments:

Reviewer's Responses to Questions

**Comments to the Author**

1. If the authors have adequately addressed your comments raised in a previous round of review and you feel that this manuscript is now acceptable for publication, you may indicate that here to bypass the “Comments to the Author” section, enter your conflict of interest statement in the “Confidential to Editor” section, and submit your "Accept" recommendation.

Reviewer #3: All comments have been addressed

2. Is the manuscript technically sound, and do the data support the conclusions?

Reviewer #3: (No Response)

3. Has the statistical analysis been performed appropriately and rigorously? 

Reviewer #3: (No Response)

4. Have the authors made all data underlying the findings in their manuscript fully available?

Reviewer #3: (No Response)

5. Is the manuscript presented in an intelligible fashion and written in standard English?

Reviewer #3: (No Response)

6. Review Comments to the Author

Reviewer #3: The review process of this manuscript is not straightforward. I appreciate the authors' perseverance in improving their manuscript.

7. PLOS authors have the option to publish the peer review history of their article (what does this mean?). If published, this will include your full peer review and any attached files.

Reviewer #3: No

---

## [Editor Report · Acceptance letter]

25 Jun 2024

PONE-D-22-30710R3 

PLOS ONE

Dear Dr. Daba, 

I'm pleased to inform you that your manuscript has been deemed suitable for publication in PLOS ONE. Congratulations! Your manuscript is now being handed over to our production team.

Kind regards, 

on behalf of

Mr. Mohammed Feyisso Shaka 

Academic Editor

PLOS ONE